# Femoral head collapse after hip intra-articular corticosteroid injection: An institutional response to improve practice and increase patient safety

Brandon J. Kelly[1], Benjamin R. Williams[1], Amy A. Gravely[2], Kersten Schwanz[3], V. Franklin Sechriest, II[1,4]*

1 Department of Orthopedic Surgery, University of Minnesota, Minneapolis, Minnesota, United States of America, 2 Department of Research Service, Veterans Affairs Medical Center, Minneapolis, Minnesota, United States of America, 3 Department of Physical Medicine/Rehabilitation, University of Minnesota, Minneapolis, Minnesota, United States of America, 4 Department of Orthopaedic Surgery, Veterans Affairs Medical Center, Minneapolis, Minnesota, United States of America

* Vernon.Sechriest@va.gov

**Data Availability Statement:** Data cannot be shared publicly because the U.S. Veterans Health Administration (VHA) places restrictions on release

## Abstract

### Introduction

Femoral head collapse (FHC) is a rarely reported complication of hip intra-articular corticosteroid injection (IACSI). Upon observing a high rate of FHC after hip IACSI, we sought to (1) describe how we addressed the problem through a systematic evaluation of clinical data and institutional care practices followed by process improvement efforts; and (2) report our results.

### Methods

Patients receiving hip IACSI during a 27-month period underwent retrospective review to determine the rate of FHC and to identify associated patient factors or practice shortfalls. Findings led to institution-wide interventions: (1) to improve patient/provider awareness of this association; and (2) to develop/implement practice guidelines. Rates of FHC after hip IACSI and practice patterns among providers before and after intervention were compared.

### Results

Initial FHC rate after hip IACSI was 20.4%. Patient-related factors included body mass index (p = 0.025), history of cancer therapy (p = 0.012), Vitamin D level (p = 0.030), and multiple injections (p = 0.004). Volume/dose of injectate and post-injection surveillance methods varied widely. Quality improvement (QI) intervention resulted in fewer treatment referrals (from 851 to 436), fewer repeat injections (mean = 1.61 to 1.37; p = 0.0006), and a 5% lower FHC rate (p = 0.1292). Variation in practice patterns persisted, so a systems-based Clinical Pathway was established.

to any non-federal entity. Data may be made available through the VHA's "DATA USE AGREEMENT FOR THE RELEASE OF DE-IDENTIFIED DATA TO A NON-FEDERAL ENTITY." The institutional point-of-contact for queries by a non-federal entity to access data for this manuscript is: Pamela Best, Privacy Officer for The Minneapolis VA Medical Center PH:(612) 467-6536 / Fax:(612) 467-2197 Email: pamela.best@va.gov.

**Funding:** The authors received no specific funding for this work.

**Competing interests:** The authors have declared that no competing interests exist.

## Discussion

When a high rate of FHC after hip IACSI was found to be associated with certain patient and practice factors, introduction of education materials and treatment guidelines decreased number of referrals, number of injections per patient, and FHC rate. In the absence of the systems-based Pathway, the type, dose, and volume of injectate and post-procedure follow-up remained variable.

## Introduction

Intra-articular corticosteroid injection (IACSI) is considered a safe and effective treatment for hip osteoarthritis (OA) [1]. Femoral head collapse (FHC) is a complication of this treatment [2–7]. However, incidence of FHC after IACSI is unknown, and risk factors are not well-defined.

From 2015 to 2017, we observed several patients with hip OA treated with IACSI who developed FHC. Given limited information on risk factors for this adverse outcome and lack of best-practices for prevention, a quality improvement (QI) investigation was undertaken to address the problem. Initial correlation with patient-related factors led to an institution-wide campaign to educate patients and providers. Discovery of variation in procedural and peri-procedural practices led to the development and introduction of treatment guidelines for providers.

We conducted a retrospective study of patients undergoing hip IACSI in our medical center before and after QI intervention. Goals of this study were (1) to identify patient risk factors or practice shortfalls associated with FHC after IACSI; and (2) to assess the impact of a QI intervention on institutional care practices and patient outcomes.

## Materials and methods

This was a single center, before/after quasi-experimental study designed to bring about immediate quality improvements in health delivery in our institution. The Institutional Review Board provided exemption for this investigation and there was no external funding.

### Quality improvement inception

After witnessing multiple cases of FHC after hip IACSI, we reviewed and compared our experience and institutional care practices with evidence-based literature. Although evidence supported IACSI for treatment of hip OA, no strong evidence of association with FHC existed. FHC was not acknowledged as a potential complication by the American Academy of Orthopaedic Surgeons (AAOS) appropriate use criteria [1]. There was no consensus as to best/safest injectate, and limited/no consensus on post-injection follow-up practices. A QI investigation was undertaken to inform risk management and practice improvement.

### Preliminary data collection and analysis

From October 1, 2015 through December 31, 2017, patients who underwent hip IACSI were identified using Radiology procedural records. Patients with artificial hips and patients injected prior to MRI-arthrography were excluded. The electronic medical record was reviewed for patient characteristics: age, sex, body mass index (BMI), hip pathology, alcohol abuse, tobacco use, chronic corticosteroid use, history of cancer therapy, diabetes mellitus,

obstructive pulmonary disease, prior hip trauma, HIV/AIDS, hip IACSI prior to study period, and number of hip IACSIs during the study period. If obtained during the study period, hemoglobin A1c, albumin, and Vitamin D levels were tabulated.

Anterior-posterior pelvis and/or hip radiographs were reviewed using the Picture Archiving and Communication System (PACS) (Visage version 7.1, Richmond, Victoria, Australia). All patients studied had pre-injection radiographs. If a patient did not have a pelvis or hip radiograph available at least one-month post-injection, the patient was excluded from FHC analysis, but was included in analysis of post-injection radiographic follow-up. All radiographs were obtained using our institution's standardized Radiology department protocol. No images were excluded for poor quality. Timing of pre- and post-injection radiographs was non-standardized. Each pre- and post-injection radiograph was compared side-by-side for interval changes. Post-injection FHC was defined radiographically as new loss of femoral head sphericity and was measured by comparing the contour of the weightbearing portion of the femoral head on each radiograph. For each hip image, a circle tool was used to outline the contour of the subchondral bone-edge in the weightbearing portion of the femoral head. Images were initially compared and interpreted by two orthopedic surgery residents and later independently verified by an attending orthopedic staff. In all cases, FHC was dramatic, easily detected, and the interim change in femoral head sphericity secondary to collapse was confirmed by official radiology reports (Fig 1).

## Quality improvement intervention

After the initial investigation, a workgroup was formed comprised of stakeholders from Orthopedics, Radiology, Physical Medicine/Rehabilitation, and Pharmacy. The work-group reviewed preliminary institutional data and evidence-based literature to develop treatment guidelines addressing: 1) indications/contraindications; 2) medications; 3) injection technique; and 4) post-injection surveillance (Fig 2). Education materials were also created (Fig 3). Next, a high-profile, well-coordinated, institution-wide, campaign to raise awareness of work-group findings and recommendations was undertaken to educate providers in Radiology, Orthopedics, Rheumatology, Physical Medicine/Rehabilitation, and Primary Care. Guidelines and

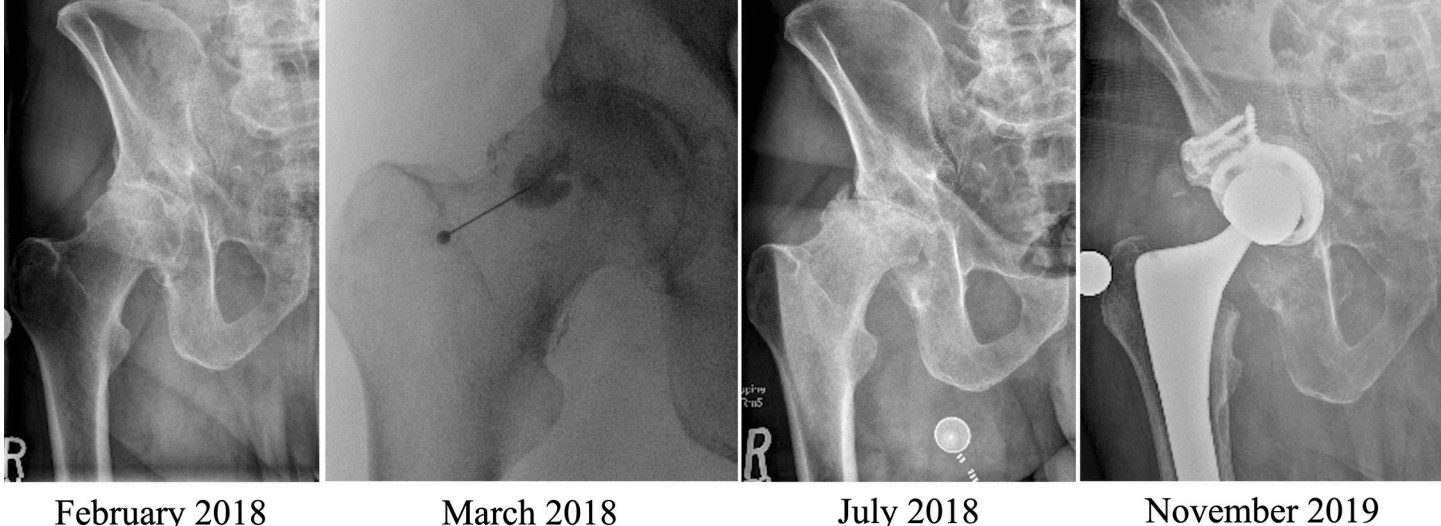

| February 2018 | March 2018 | July 2018 | November 2019 |

**Fig 1. FHC after hip IACSI.** 72-year-old male with symptomatic right hip OA. Progression from moderate/severe radiographic hip OA to FHC after first-time hip IACSI. Patient subsequently had THA complicated by superior and posterior acetabulum bone loss necessitating use of an augmented acetabulum component and bone graft.

# Initial Quality Improvement Hip IASCI Work-Group Recommendations

1. <u>Indications</u>
   a) Inflammatory/non-inflammatory hip arthropathy
2. <u>Contraindications</u>
   a) <u>Absolute:</u>
      i. Femoral head avascular necrosis
      ii. Pre-existing FHC
      iii. Active infection
      iv. Intracapsular hip fracture without evidence of bony union
   b) <u>Relative:</u>
      i. Osteoporosis or low bone mineral density
      ii. Advanced hip OA with cystic changes in femoral head on radiographs
3. <u>Medication Mixture</u>
   a) Methylprednisolone acetate or dexamethasone
   b) Ropivacaine hydrochloride 0.5%
4. <u>Injection Technique</u>
   a) Image-guided (fluoroscopy or ultrasound)
5. <u>Post-Injection Surveillance</u>
   a) Followup examination and hip radiographs 3-6 months after injection

**Fig 2. Initial informal hip IACSI treatment guidelines derived by multidisciplinary work-group.**

patient-education materials were then distributed hospital-wide for their use in clinical practice. Work-group findings and practice recommendations were also highlighted at our institution's annual research symposium. Over the next 27 months, the number of hip IACSI referrals, number of injections per hip, FHC rate, and post-injection radiograph rate were tabulated and compared to patients treated pre-intervention.

## Statistical analysis

Descriptive statistics were used to summarize patient characteristics (Table 1). To examine bivariate relationships between modifiable and non-modifiable patient-predictors and FHC, comparisons between patients with and without FHC were made. Pearson's Chi-Square test was used for discrete variables, while two sample t-test was used for continuous variables. Statistical analysis was performed using SAS version 9.4 (Cary, North Carolina). P-values were considered statistically significant if $< 0.05$.

## Results

### Initial quality improvement findings

**Patient factors.** From October 1, 2015 through December 31, 2017, 851 hip IACSIs were performed on 531 hips (458 patients). Mean number of IACSIs per hip = 1.61 ± 1.06. Seventy-four patients (13.9%) had an ipsilateral hip IACSI prior to this study period. The majority of patients were male [502 (94.5%) vs 29 (5.5%)]. There were 294 right hips (55.4%) and 237 left hips (44.6%). Mean age = 67.8 years ± 11.7 and mean BMI = 31.4 kg/m$^2$ ± 5.7. Diagnoses included OA (93.4%), femoroacetabular impingement (FAI) (2.1%), avascular necrosis (AVN)

# Hip Cortisone Injection

## Why is my provider recommending a cortisone injection for my hip?

- This treatment involves injection of steroid medication and local anesthetic into the hip joint. It is intended to reduce pain and inflammation and is just one part of your overall treatment plan. You may also need physical therapy, a weight loss plan, medications, etc.

- The injection may also be used to differentiate hip pain from other areas, especially the spine or nerves in the back and leg.

## Who performs my hip cortisone injection?

- A radiologist or physiatrist performs the hip injection under image guidance (i.e. ultrasound or fluoroscopy) to ensure the medicine is delivered to the right location.

## What are the risks or side effects with a cortisone injection in the hip?

- **Pain and swelling.** Common for 1 to 2 days after the injection. Rest and ice may be helpful.

- **Skin discoloration.** At the injection site, the skin may lighten in color and there may be thinning of the fat layer under the skin.

- **Elevated blood sugar.** If you have diabetes, you may have temporary change in blood sugar control. If blood sugar levels remain higher than normal, contact your diabetes provider.

- **Infection.** It is possible for the hip to become infected. This is rare. If you suspect an infection or run a fever (101.3 ° F or higher), contact your provider immediately.

- **Allergic reaction.** Some patients have allergic reactions to the local anesthetic added to the injection. Allergic reactions to cortisone are rare. Cortisone is a synthetic version of cortisol, a steroid naturally found in the body. If you suspect a reaction, contact your provider or seek emergency medical attention if needed.

- **Rapid progression of hip arthritis.** For unknown reasons, some patients experience worsening of their hip condition. After the injection, if you have increased hip pain or worsening hip symptoms, contact your provider and schedule a follow up visit.

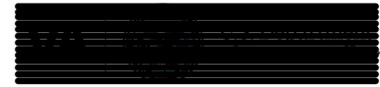

**Fig 3. Procedure-specific education material available for all patients and providers at our institution.**

**Table 1. Hip IACSI patient demographics and medical profiles.**

| | |
|---|---|
| Hip IACSI | 851 |
| Patients | 458 |
| Hips | 531 |
| Mean Hip IACSI During Study | 1.61 ± 1.06 |
| Hip IACSI Prior to Study Dates | |
| Yes | 74 (13.94%) |
| No | 457 (86.06%) |
| Age (years) | 67.77 ± 11.72 |
| Sex | |
| Male | 502 (94.54%) |
| Female | 29 (5.46%) |
| BMI (kg/m$^2$) | 31.42 ± 5.71 |
| Hip Laterality | |
| Right | 294 (55.37%) |
| Left | 237 (44.63%) |
| Diagnosis | |
| Hip OA | 496 (93.41%) |
| FAI | 11 (2.07%) |
| AVN | 8 (1.51%) |
| Labral Tear | 8 (1.51%) |
| Hip Pain | 4 (0.75%) |
| Hip Dysplasia | 2 (0.38%) |
| Psoriatic Arthritis | 2 (0.38%) |
| Alcohol Abuse | |
| Yes | 92 (17.33%) |
| No | 439 (82.67%) |
| Tobacco Use | |
| Yes | 170 (32.02%) |
| No | 361 (67.98%) |
| Chronic Steroid Use | |
| Yes | 24 (4.52%) |
| No | 507 (95.48%) |
| History of Hip Trauma | |
| Yes | 14 (2.64%) |
| No | 517 (97.36%) |
| History of Chemotherapy or Radiation Therapy | |
| Yes | 44 (8.29%) |
| No | 487 (91.71%) |
| Obstructive Pulmonary Disease | |
| Yes | 80 (15.07%) |
| No | 487 (84.93%) |
| Diabetes Mellitus | |
| Yes | 149 (28.06%) |
| No | 382 (71.94%) |
| Serum Albumin (g/dL, reference range 3.5–5.2) | 3.71 ± 0.45 |
| Hemoglobin A1c (%, reference range 4.0–6.0) | 6.08 ± 1.24 |

(*Continued*)

**Table 1.** (Continued)

| | |
|---|---|
| Serum Vitamin D (ng/mL, reference range 20–50) | 26.21 ± 11.61 |

*Results are reported as the mean and standard deviation for continuous variables and the count and percentage for discrete variables.

**HIV/AIDS was not a significant variable in our analysis and was omitted from final data reporting in order to protect sensitive patient information.

(1.5%), labral tear (1.5%), unidentified hip pain (0.75%), dysplasia (0.38%), and psoriatic arthritis (0.38%) (Table 1).

Of 531 hips, 412 (77.6%) in 360 patients met inclusion criteria. Of 119 hips excluded, 75 hips (63.0%) lacked post-IACSI radiographs and 44 (37.0%) underwent THA prior to post-IACSI radiographs. To account for non-responder bias, patients excluded were compared to patients included in the analysis. Excluded patients were similar except for a higher mean hemoglobin A1c (6.41% vs. 5.99%, p = 0.012), lower number of injections per hip (1.44 vs. 1.67, p = 0.015), and a higher percentage of HIV/AIDS diagnoses (final numeric reporting was omitted to protect sensitive patient information).

Of 412 hips in the final analysis, 20.4% (84/412) developed FHC. Patients with FHC had a higher mean BMI (33.0 kg/m$^2$ vs. 31.1 kg/m$^2$, p = 0.025), higher percentage of cancer therapy (14.3% vs. 6.1%, p = 0.012), lower serum Vitamin D levels (23.1 ng/mL vs. 26.8 ng/mL, p = 0.030), and lower hemoglobin A1c levels (5.58% vs. 6.09%, p < 0.001). History of prior hip IACSI, age, sex, laterality, alcohol abuse, tobacco use, chronic steroid use, diabetes mellitus, hip trauma, HIV/AIDS, obstructive pulmonary disease, and serum albumin level were not significant. Hips with FHC underwent more IACSIs during the study period [2.05 (0.91/year) vs. 1.57 (0.70/year), p = 0.004]. There were 8 hips (1.51%) that had a diagnosis of avascular necrosis (AVN). One patient was excluded due to pre-existing FHC. Of the remaining 7 hips, 2 hips (2.38%) developed FHC, and 5 hips (1.52%) did not develop collapse after hip IACSI. (Table 2).

While the rate of FHC after IACSI for patients excluded from our pre-intervention analysis is unknown, a best-case scenario would be that none developed collapse during the study period based on lack of need for radiographic follow-up. If this were the case, the initial FHC rate would be 15.8% (84/531).

**Practice-related factors.** Specialties that referred patients for hip IACSI included Orthopedics, Rheumatology, Physical Medicine/Rehabilitation, and Primary Care. Procedure-specific education materials did not exist. All hips were injected under fluoroscopy with 0.5% ropivacaine hydrochloride (range 3.0–6.0 mL), triamcinolone acetonide (range 1–2 mL; 40 mg/mL), and iopromide contrast (1 mL). Medication volume/dose were at discretion of the radiologist. No significant association was identified between FHC and local anesthetic or corticosteroid dosage, volume, or medication type. Post-procedure, patient follow-up with the referring provider varied. In most cases, decision for follow-up was left to the patient. Post-IACSI radiographs were ordered/obtained for only 77.6% of patients. Patients frequently underwent repeat injections without interval office visits or new radiographs.

## Quality improvement intervention impact

In the 27 months after QI investigation and implementation of practice recommendations (Fig 2), reduction in number of procedures was 48.8% (851 to 436), reduction in number of hips injected was 39.6% (531 to 321), number of injections per hip reduced from 1.61 to 1.37 (p = 0.0006), reduction in post-IACSI FHC was 67.9% (84 to 27), and rate of FHC decreased

**Table 2. Comparison of patient demographics and medical profiles: FHC vs. no FHC after hip IACSI.**

| Collapse | | YES | NO | P-Value |
|---|---|---|---|---|
| | | N = 84 | N = 328 | |
| | | (20.4%) | (79.6%) | |
| Hip IACSI | | 172 | 515 | n/a |
| Patients | | 82 | 284 | n/a |
| Hips | | 84 | 328 | n/a |
| Mean Hip IACSI During Study | | 2.05 ± 1.40 | 1.57 ±1.01 | **0.0040[b]** |
| Hip IACSI Prior to Study Dates | | | | 0.7481[a] |
| | Yes | 14 (16.67%) | 50 (15.24%) | |
| | No | 70 (83.33%) | 278 (84.76%) | |
| Age (Years) | | 66.33 ±10.95 | 67.62 ±12.25 | 0.7481[b] |
| Sex | | | | 0.5634[a] |
| | Male | 78 (92.86%) | 310 (94.51%) | |
| | Female | 6 (7.14%) | 18 (5.49%) | |
| BMI | | 33.00 ±7.12 | 31.09 ±5.46 | **0.0245[b]** |
| Hip Laterality | | | | 0.2313[a] |
| | Right | 41 (48.81%) | 184 (56.1%) | |
| | Left | 43 (51.19%) | 144 (43.9%) | |
| Diagnosis | | | | 0.6083[a] |
| | Hip OA | 81 (96.43%) | 301 (91.77%) | |
| | FAI | 1 (1.19%) | 8 (2.44%) | |
| | AVN | 2 (2.38%) | 5 (1.52%) | |
| | Labral Tear | 0 (0%) | 6 (1.83%) | |
| | Hip Pain | 0 (0%) | 4 (1.22%) | |
| | Hip Dysplasia | 0 (0%) | 2 (0.61%) | |
| | Psoriatic Arthritis | 0 (0%) | 2 (0.61%) | |
| Alcohol Abuse | | | | 0.7781[a] |
| | Yes | 15 (17.86%) | 63 (19.21%) | |
| | No | 69 (82.14%) | 265 (80.79%) | |
| Tobacco Abuse | | | | 0.7346[a] |
| | Yes | 28 (33.33%) | 103 (31.4%) | |
| | No | 56 (66.67%) | 225 (68.6%) | |
| Chronic Steroid Use | | | | 0.6896[a] |
| | Yes | 5 (5.95%) | 16 (4.88%) | |
| | No | 79 (94.05%) | 312 (95.12%) | |
| History of Hip Trauma | | | | 0.6874[a] |
| | Yes | 3 (3.57%) | 9 (2.74%) | |
| | No | 81 (96.43%) | 319 (97.26%) | |
| History of Chemotherapy or Radiation Therapy | | | | **0.0124[a]** |
| | Yes | 12 (14.29%) | 20 (6.1%) | |
| | No | 72 (85.71%) | 308 (93.9%) | |
| Obstructive Pulmonary Disease | | | | 0.6690[a] |
| | Yes | 11 (13.1%) | 49 (14.94%) | |
| | No | 73 (86.9%) | 279 (85.06%) | |
| Diabetes Mellitus | | | | 0.0537[a] |
| | Yes | 16 (19.05%) | 97 (29.57%) | |
| | No | 68 (80.95%) | 231 (70.43%) | |

(*Continued*)

**Table 2.** (Continued)

| Collapse | YES | NO | P-Value |
|---|---|---|---|
| | N = 84 | N = 328 | |
| | (20.4%) | (79.6%) | |
| Serum Albumin | 3.67 (± 0.43) | 3.74 (±0.45) | 0.2169[b] |
| (g/dL, reference range 3.5–5.2) | | | |
| Hemoglobin A1c | 5.58 (± 0.60) | 6.09 (± 1.22) | <**0.0001**[b] |
| (%, reference range 4.0–6.0) | | | |
| Serum Vitamin D (ng/mL, reference range 20–50) | 23.14 (±11.25) | 26.84 (11.28) | **0.0301**[b] |

[a]Pearson's Chi-Square Test.

[b]Two Sample Test.

Significance was set at p < 0.05.

*Results are reported as the mean and standard deviation for continuous variables and the count and percentage for discrete variables.

**HIV/AIDS was not a significant variable in our analysis and was omitted from final data reporting in order to protect sensitive patient information.

from 20.4% to 15.1% (p = 0.1292) (Table 3). Variation in medication volume/dose persisted. Variation in radiographic follow-up requirement persisted as x-rays were obtained for only 55.8% of patients post-procedure.

While the rate of FHC after IACSI for patients without post-injection radiographs is unknown, a best-case scenario would be that none developed collapse during the study period. If this were the case, the post-QI intervention FHC rate would be 8.4% (27/321).

## Development of a clinical pathway for hip IACSI

During the investigation/intervention, the work-group was developing a systems-based Clinical Pathway for hip IACSI (Fig 4) to expand on the initial informal recommendations (Fig 2). Based on lessons-learned, the work-group created a flowchart to guide peri-procedural care. Through a series of meetings and reviews, this flowchart was revised to incorporate as many evidence-based care practices as possible. When evidence-based practices were not available, practices were adopted on the basis of community standards and/or consensus among work-group members. In addition to mapping out the sequence of patient care, the Pathway incorporated use of patient-education materials, standardized physician order-sets for medications and follow-up, and pharmacy involvement/oversight.

**Table 3. Hip IACSI QI impact.**

| | Pre-QI | Post-QI | P-Value |
|---|---|---|---|
| | 10/1/2015-12/31/2017 | 1/1/2018-3/11/2020 | |
| Hip IACSI | 851 | 436 | n/a |
| Patients | 458 | 280 | n/a |
| Hips | 531 | 321 | n/a |
| Mean Hip IACSI | 1.67 ± 1.12 | 1.37 ± 0.84 | **0.0006**[b] |
| Femoral Head Collapse Number | 84 | 27 | n/a |
| Femoral Head Collapse Percentage | 20.4% | 15.1% | **0.1292**[a] |

[a]Pearson's Chi-Square Test.

[b]Two Sample Test.

Significance was set at p < 0.05.

*Results are reported as the mean and standard deviation for continuous variables and the count and percentage for discrete variables.

## Hip Intra-Articular Corticosteroid Injection Clinical Pathway

**Outpatient "Hip Pain" Consultation**
- H&P
- Hip X-Rays
- Treatment options reviewed & patient documented to have inadequate response to non-pharmacologic conservative measures and simple analgesics
- Hip IACSI indications/contraindications reviewed
- Standardized education material reviewed with patient with shared decision making
- Hip IACSI ordered

**Hip IACSI Order Set**
- Radiology referral for scheduling
- Pharmacy referral for review & medication preparation
- Schedule follow-up visit + X-rays (~3 months after injection)

**Radiology Review and Scheduling**
- Radiology staff review request & confirm patient meets clinical criteria
- Patient scheduled

**Pharmacy Review and Medication Preparation**
- Pharmacy staff review request and confirm no contraindications to medications (i.e. allergy)
- On date of procedure, prepare standardized medication mixture
- Record laterality-specific medication information into the medication administration record

**Hip IACSI Administration**
- On date of procedure, Radiology staff place medication order for hip IACSI
- Pharmacy fulfills order for standardized medication mixture
- Radiology staff use "Hip IACSI Patient Education" form to augment informed consent process
- Hip IACSI administered under image guidance

**Post-Hip IACSI Surveillance**
- Immediately post: Radiologist confirms patient has 3-month followup appointment + X-rays with referring provider
- 24 hours post: Radiology RN contacts patients to assess symptoms and need for earlier followup visit with referring provider

**Hip IACSI Indications and Contraindications**
Indications:
- Non inflammatory hip arthropathy
- Inflammatory hip arthropathy
- Palliative (hip pathology other than infection where surgery is contraindicated)

Absolute Contraindications:
- Active infection
- Hypersensitivity to intra-articular agent
- Unstable coagulopathy
- Poorly controlled diabetes mellitus (i.e. Hemoglobin A1c > 9.0) and/or poor glycemic control (persistently elevated blood glucose: range from 200-500 mg/dL)
- Femoral head or neck fracture without evidence of healing

Relative Contraindications:
- Metabolic disorder with low bone mineral density
- Morbid obesity (BMI > 40 kg/m$^2$)
- History of chemotherapy or radiation therapy
- Avascular necrosis of femoral head
- Pre-existing femoral head collapse
- Advanced hip OA with subchondral hypomineralization and cyst formation
- Presence of prosthetic joint

**Standardized Medication for Injection**
- 20 mg Methylprednisolone acetate (1 mL)
- 0.5% Ropivacaine hydrochloride (3 mL)

**Fig 4. System-wide clinical pathway for hip IACSI.**

## Discussion

FHC after hip IACSI has been described in case reports and small case series. Suggested etiologies include toxicity from injectate, volume-related increased intracapsular pressure restricting femoral head blood supply, and occult subchondral fracture or AVN progressing to collapse with medication-related inhibition of joint pain [2–7]. While no cause-effect relationship is known, the association between such a common treatment and such a pathogenic process is concerning.

At the inception of this investigation, we had already managed several patients with FHC occurring shortly after hip IACSI. Cases were characterized by acute increase in hip pain and disability, radiographically-confirmed articular collapse, and adjustment in care plan to allow earliest possible total hip arthroplasty (THA). A small number of patients with FHC less than 3 months post-IACSI required delay in THA to decrease risk of prosthetic joint infection [8]. Consequently, their care was further complicated by an increase in sedentary behavior and presentation for surgery in a deconditioned state (Fig 5).

Initial QI analysis of FHC after IACSI suggested certain patient factors were associated (elevated BMI, history of cancer therapy, low Vitamin D level). Our analysis also revealed lack of consistency in volume/dose of injectate, time-interval between treatments, and post-injection follow-up. The practice inconsistencies permitted many patients to receive hip injections "on demand", often without pre-injection clinical or radiographic evaluation. As our analysis also suggested that multiple IACSIs within a narrow time-frame (27 months) were associated with FHC, we recognized the need for improved organization/oversight.

To improve safety and consistency of hip IACSI, we created procedure-specific education materials. We also developed guidelines for procedural and peri-procedural practices and

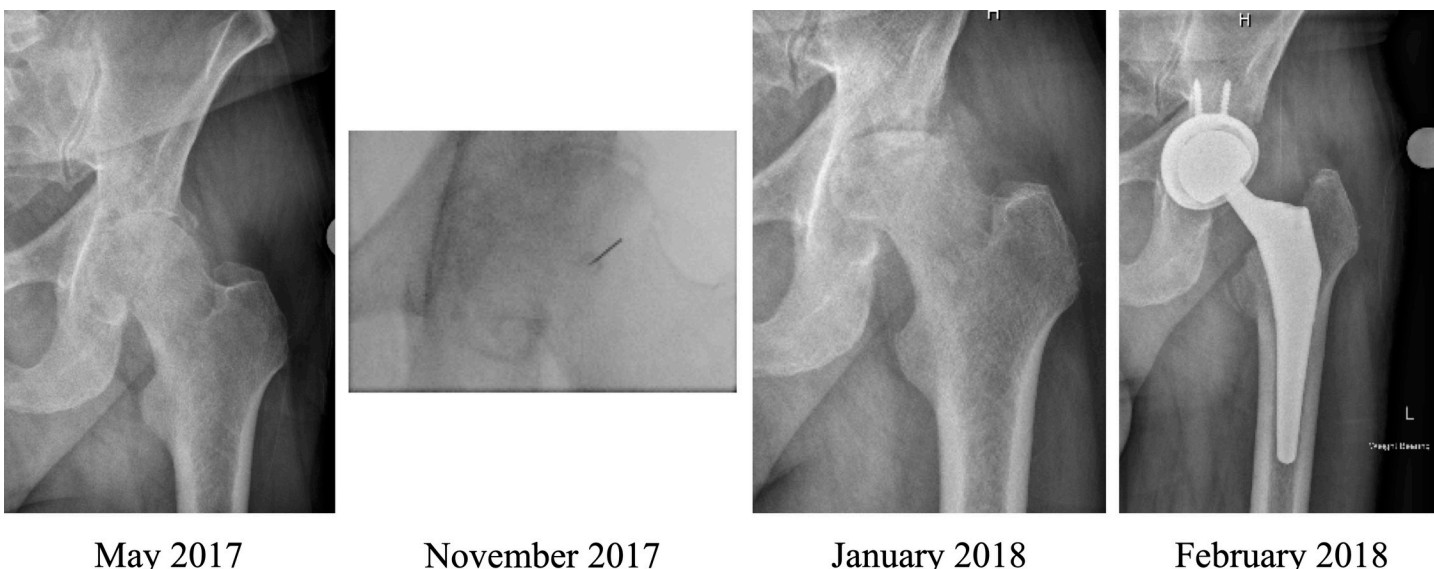

<div style="text-align:center">

May 2017   November 2017   January 2018   February 2018

</div>

**Fig 5. Rapid progression to FHC less than three months after hip IACSI in 66-year-old male with left hip OA.** After FHC, patient developed significant quad weakness and was unable to ambulate during January 2018 office visit. Left THA was delayed until three months after hip IACSI to reduce risk of PJI.

distributed to all referring and treating providers. Importantly, given the urgent need to improve patient safety, our initial intervention relied entirely on individual providers' awareness and responsibility. The systems-based Clinical Pathway was in development (Fig 4), but not yet in place.

In the two years that followed the initial intervention, hip IACSIs decreased (approximately half as many) along with fewer repeat treatments. While it is not certain that these referral patterns resulted directly from providers following our recommendations, the rate of FHC after IACSI decreased by 5%. However, in the absence of the systems-based Clinical Pathway, volume/dose of injectate and post-injection follow-up remained variable.

### Patient factors

**Obesity.** Patients who developed FHC after IACSI had higher mean BMIs (33.00 kg/m$^2$ vs. 31.09 kg/m$^2$, p = 0.025). Only one other study has analyzed BMI as a variable for FHC, and no association was noted [5]. Yet, obesity is associated with AVN and subchondral insufficiency fracture [9–11]. Such conditions may account for some cases of FHC in our analysis. Ultimately, as obese patients have increased risk for hip conditions characterized by unhealthy bone, treatments that decrease joint pain and may promote increased joint stress (e.g. IACSI) may also increase risk for FHC. For our obese patients, the decision for IACSI is always informed by complete clinical history and recent hip imaging.

**History of treatment for malignancy.** Patients developing FHC after IACSI had a higher percentage history of cancer therapy (14.3% vs. 6.1%, p = 0.012). Radiation and chemotherapy are linked to bone damage/loss and increased fracture risk [12, 13]. This association has important implications for palliative care of hip pain. For patients undergoing cancer therapy, THA may be contraindicated and hip IACSI preferred for symptomatic relief. We conclude that such patients should be selected carefully and followed closely if IACSI performed.

**Low serum Vitamin D levels.** Patients developing FHC after IACSI had lower serum Vitamin D levels. However, our finding must be interpreted with caution as these levels were not tested at standardized intervals around the time of each hip procedure. Furthermore, the

role of Vitamin D levels in the evaluation and treatment of orthopaedic patients remains controversial, and supplementation therapy has not been found to prevent osteoporosis-related fractures [14, 15]. Ultimately, while the association is logical, no firm conclusion can be drawn from our analysis.

**Avascular necrosis.** The natural history of avascular necrosis of the femoral head is FHC. In our analysis there were only 8 hips (1.51%) that had a diagnosis of AVN (7 ultimately included in analysis). Of the patients with AVN there was a 28.57% FHC rate (2/7) after IACSI. While there is a possibility AVN was underdiagnosed in our patient population, we were limited by the retrospective nature of the review and the information contained in the medical record at the time of retrieval. Due to the slightly higher FHC rate seen with this diagnosis and the natural history of the disease, we consider AVN as a relative contraindication for hip IACSI.

## Practice-factors

**Volume/dose of medications injected.** Initial investigation demonstrated variation in dose of triamcinolone acetonide (range, 40–80 mg/mL), volume of ropivacaine hydrochloride (range 3.0–6.0 mL), and total volume of injectate (range, 5–9 mL). We endeavored to research and recommend the best/safest medication mixture for our patients. This was challenging as there is no consensus as to what constitutes "best practice" for hip injectate.

Regarding medications, our literature review suggested all local anesthetics and corticosteroids have chondrotoxic effects, but that methylprednisolone acetate, dexamethasone, and ropivacaine hydrochloride were the least chondrotoxic [16–23]. With regard to total volume of injectate, literature review confirmed increased intra-capsular pressure may cause hip pathology (i.e. AVN) and provided insight into the relationship between volume of injection and intracapsular hip joint pressure [24]. To minimize risk related to increased intra-capsular pressure, we recommended injecting the lowest possible volume, $\leq$ 5mL [25] (Fig 6).

Final recommendations for medications were derived through collaboration with all clinical stakeholders. However, at the time of this study these recommendations were not yet part of the formal Clinical Pathway with standardized order-sets and pharmacy involvement/oversight. Despite efforts to encourage more consistent practice, without the system-based Pathway yet in place, the type, dose, and volume of medications injected remained highly variable.

## Patients receiving multiple injections / lack of consistent follow-up after procedure

Patients treated with greater than one hip IACSI during the initial 27-month study period were more likely to develop FHC. While the number and frequency of hip IACSIs has never been reported as a risk factor for FHC, it is logical that repeated injections of medications with dose- and time-dependent toxicity in a narrow timeframe might increase risk of joint damage [16–23].

Initial investigation revealed repeat IACSIs were commonly ordered at the patient's request, oftentimes without pre-procedure clinical or radiographic evaluation. A small number of patients were undergoing regularly scheduled hip IACSIs (i.e. every three or six months). While all patients who underwent first-time hip IACSI had pre-procedure hip radiographs, there was no consistent practice for post-IACSI imaging or imaging between procedures. Initial findings were that post-injection radiographs were obtained for 77.6% of patients. Without strong evidence to inform the ideal frequency of hip IACSIs and/or best follow-up practices, work-group consensus was to adopt a very conservative approach. Specifically, prior to referral for initial or repeat IACSI, guidelines recommended in-person evaluation with new radiographs to assess the hip and to review/discuss treatment alternatives.

| | Local Anesthetic | Corticosteroid | Contrast Dye |
|---|---|---|---|
| **Medication** | 0.5% Ropivacaine Hydrochloride | 20 mg Methylprednisolone Acetate<br>*or*<br>4 mg Dexamethasone Phosphate<br>*or*<br>8 mg Dexamethasone | Iopromide |
| **Recommended Volume** | 1-3 mL | 1 mL | 1 mL |

**Fig 6. Hip IACSI medication recommendations based on best-available evidence literature review and local/community standards.**

After QI intervention, we observed a significant decrease in number of repeat injections per patient, mean = 1.61 vs. 1.37, (p = 0.0006). However, variation persisted in follow-up practices and the number of patients with post-injection radiographs decreased to 55.8% from 77.6%. Many patients continued to receive one or more injections between hip radiographs. Again, this persistent variation is attributed to lack of the systems-based Pathway during the study period and reliance on individual providers to follow work-group guidelines.

### Strengths

To our knowledge, our experience with identifying and managing FHC after hip IACSI is the largest series ever reported. Our investigation was performed at a single institution in a single health system. The electronic medical record and integrated PACS facilitated complete and consistent data collection. Our study demonstrates how a QI investigation and intervention may improve patient safety. Our study also demonstrates the formidable challenges associated with implementing sustainable institution-wide change.

### Limitations

The primary study limitation is its retrospective observational design comparing nonrandomized groups. A randomized controlled trial would better define the relationship between hip IACSI and FHC and/or establish any causality related to patient, provider, or environment. Although our initial QI investigation identified potential patient-related risk factors for FHC after IACSI and made recommendations for/against treatment based on these, the impact of our QI intervention on number of patients referred for hip IACSI with potential risk factors was not tracked or tabulated. Also, the unique population studied (predominantly older males) may limit the generalizability of findings and recommendations. It is also possible that all or some of the FHC observed in this study occurred as part of the natural progression of

patients' hip OA. However, to our knowledge, there is no natural history study that reports a similar or accepted rate of FHC in patients with hip OA.

Due to the retrospective nature of this work, the radiographic follow-up for pre- and post-intervention study groups was non-standardized. Additionally, the time to chart/radiographic review was shorter for the post-intervention group, thereby introducing the potential for a falsely decreased FHC rate in the post-intervention group. While the shorter length of radiographic follow up for the post-intervention group and the possibility of a falsely low FHC rate in this group is a weakness of our study, it strengthens our rationale for instituting the Clinical Pathway to standardize hip IACSI practices so to allow for further prospective study.

Finally, a large number of patients (22.4% pre-intervention and 44.2% post-intervention) were excluded from final analysis of occurrence of FHC due to lack of post-injection imaging. However, even if all patients initially excluded due to incomplete radiographic follow-up were included in the final analysis with the assumption of "no collapse", FHC rates after IACSI remain high (i.e. pre-intervention = 15.8%; post-intervention = 8.4%). At our institution, even this "best case scenario" was sufficiently concerning to motivate further investigation and intervention.

## Conclusion

When a QI investigation suggested FHC after hip IACSI was associated with certain patient and practice factors, we rapidly intervened with introduction of new education materials and treatment guidelines. This intervention decreased the number of hip injection referrals, number of repeat injections, and the rate of post-injection FHC. However, the intervention failed to improve the consistency of medications injected and post-procedure follow-up practices. The failure to change key practice patterns is attributed to the absence of the systems-based Clinical Pathway during the study period. Although the Pathway was in development during this time, progress was slow due to the need to involve multiple stakeholders in decision-making, time required to create standardized order-sets in the electronic medical record, and the lengthy process of gaining acceptance of a new workflow. As we were confronted with an immediate need to address a patient safety issue, our initial intervention that relied on patient/provider awareness and responsibility was all that was possible.

As a next step, to further improve quality, safety, and consistency of the practice of hip IACSI at our institution, we have introduced a formal systems-based Clinical Pathway for hip IACSI. At a minimum, standardization of this care practice should simplify future risk management. Future research will show if this Pathway improves clinical outcomes and/or further decreases rates of FHC.

## Author Contributions

**Conceptualization:** Brandon J. Kelly, Benjamin R. Williams, Amy A. Gravely, V. Franklin Sechriest, II.

**Data curation:** Brandon J. Kelly, Benjamin R. Williams, Amy A. Gravely, Kersten Schwanz, V. Franklin Sechriest, II.

**Formal analysis:** Brandon J. Kelly, Amy A. Gravely, V. Franklin Sechriest, II.

**Investigation:** Brandon J. Kelly, V. Franklin Sechriest, II.

**Methodology:** Brandon J. Kelly, Amy A. Gravely, V. Franklin Sechriest, II.

**Project administration:** V. Franklin Sechriest, II.

**Resources:** V. Franklin Sechriest, II.

**Supervision:** V. Franklin Sechriest, II.

**Validation:** Amy A. Gravely, V. Franklin Sechriest, II.

**Writing – original draft:** Brandon J. Kelly, Benjamin R. Williams, Amy A. Gravely, V. Franklin Sechriest, II.

**Writing – review & editing:** Brandon J. Kelly, Amy A. Gravely, V. Franklin Sechriest, II.

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
