## [Decision Letter · Decision Letter 0]

8 Apr 2021

PONE-D-21-00877

Femoral Head Collapse After Hip Intra-Articular Corticosteroid Injection: An Institutional Response to Improve Practice and Increase Patient Safety

PLOS ONE

Dear Dr. Sechriest,

Thank you for submitting your manuscript to PLOS ONE. After careful consideration, we feel that it has merit but does not fully meet PLOS ONE’s publication criteria as it currently stands. Therefore, we invite you to submit a revised version of the manuscript that addresses the points raised during the review process.

The authors are required to respond to all reviewers' comments and concerns especially the low rate of postinjection follow-up radiographs in the intervention group. Only 55.8% received a post-procedure x-ray of the hip. How did the authors exclude the development of FHC without follow-up radiographs? How did the authors accept this while the missing of a postinjection X-ray was an exclusion criteria.Please submit your revised manuscript by May 23 2021 11:59PM. If you will need more time than this to complete your revisions, please reply to this message or contact the journal office at plosone@plos.org. Please include the following items when submitting your revised manuscript:
A rebuttal letter that responds to each point raised by the academic editor and reviewer(s). You should upload this letter as a separate file labeled 'Response to Reviewers'.A marked-up copy of your manuscript that highlights changes made to the original version. You should upload this as a separate file labeled 'Revised Manuscript with Track Changes'.An unmarked version of your revised paper without tracked changes. You should upload this as a separate file labeled 'Manuscript'.

We look forward to receiving your revised manuscript.

Kind regards,

Osama Farouk

Academic Editor

PLOS ONE

Journal Requirements:

Reviewers' comments:

Reviewer's Responses to Questions

**Comments to the Author**

1. Is the manuscript technically sound, and do the data support the conclusions?

Reviewer #1: No

Reviewer #2: Yes

Reviewer #3: No

2. Has the statistical analysis been performed appropriately and rigorously? 

Reviewer #1: Yes

Reviewer #2: Yes

Reviewer #3: Yes

3. Have the authors made all data underlying the findings in their manuscript fully available?

Reviewer #1: No

Reviewer #2: Yes

Reviewer #3: No

4. Is the manuscript presented in an intelligible fashion and written in standard English?

Reviewer #1: Yes

Reviewer #2: Yes

Reviewer #3: Yes

5. Review Comments to the Author

Reviewer #1: Within this study the authors analyse the impact of a treatment algorism for intraarticular corticosteroid injection (IACSI) to reduce the number of femoral head collapse (FHC). Before the implementation of the algorithm the FHC rate after IACSI was 20.4% (84/412). The authors were able to calculate independent risk factors for the development of FHC (e.g. obesity or low serum Vitamin D levels) in the first group before the intervention.

The intervention consisted in a hospital guideline addressing: 1. indication; 2. medication; 3. injection-technique and 4. post-injection surveillance

As a result the authors show that there was a reduction in the number of injected patients down to 39.6%. The number of injections per hip was reduced at the same time from 1.6 to 1.3. and as significant result the rate of FHC decreased from 20.4% to 15.1%.

Critical comment

The authors are able to show that the amount of IACSI was significantly reduced after implementation of the treatment algorithm (851:436) in a comparable period. Unfortunately the reasons for the reduction of the number of interventions is not described. It is not shown if there was a reduction of number of patients with the mentioned comorbidities in the second group.

Another major concern is the low rate of postinjection follow-up radiographs in the intervention group. Only 55.8% received a post-procedure x-ray of the hip. How did the authors exclude the development of FHC without follow-up radiographs?

This is surprising, since the missing of a postinjection X-ray was an exclusion criteria in the first group.

FHC is not always easy to detect in plane radiographs in patients with osteoarthritis of the same hip. In order to have objective results a group of independent specialists (radiologists or orthopaedics) should have analysed the radiographs.

Reviewer #2: This is the largest study of this kind, looking at the incidence of Femoral Head Collapse after Intraarticular Steroid Injection. Although retrospective in nature, this provides a basis for future study in this particular population. The data does support the conclusions. The statistical analysis was appropriate and rigorous. The data was available for review. The manner in which this manuscript is written flows well and is easily understandable.

Reviewer #3: Thank you for your paper, it represent a hard and continuous effort to improve practice, though I think it may not represent a clear research topic rather than documenting your internal audit of your practice

6. PLOS authors have the option to publish the peer review history of their article (what does this mean?). If published, this will include your full peer review and any attached files.

Reviewer #1: No

Reviewer #2: No

Reviewer #3: No

---

## [Author Response · Author response to Decision Letter 0]

12 Jun 2021

Response to Reviewer Comments

PONE-D-21-00877

Editor: 

1) Only 55.8% received a post-procedure x-ray of the hip. How did the authors exclude the development of FHC without follow-up radiographs? How did the authors accept this while the missing of a post injection X-ray was an exclusion criteria?

Author Response (part a): A goal of this study was to identify variation in practice of obtaining post-procedure X-rays of the injected hip before and after quality improvement (QI) intervention. For patients without post-procedure X-rays, we were unable to conclude if they experienced FHC, so they were excluded from the FHC analysis only. However, we included these patients in our analysis of radiographic follow-up and have reported rates of radiographic follow-up to illustrate variation in post-injection X-ray practices. We have clarified our exclusion criteria as below:

• “If a patient did not have a hip radiograph available at least one-month post-injection, the patient was excluded from FHC analysis, but was included in analysis of post-injection radiographic follow-up.” (Materials and Methods, lines 112-114)

Author Response (part b): While we cannot determine if patients without post-injection radiographs developed FHC during the study period, we can illustrate variation in practice of radiographic follow-up post-procedure. This point has been further clarified in the manuscript as below:

• Variation in radiographic follow-up requirement persisted as x-rays were obtained for only 55.8% of patients post-procedure.” (Results, lines 191 – 192).

• “After QI intervention, we observed a significant decrease in number of repeat injections per patient, mean = 1.61 vs. 1.37, (p = 0.0006). However, variation persisted in follow-up practices and the number of patients with post-injection radiographs decreased to 55.8% from 77.6%.” (Discussion, lines 305 – 308).

• “Finally, a large number of patients (22.4% pre-intervention and 44.2% post-intervention) were excluded from final analysis of occurrence of FHC due to lack of post-injection imaging.” (Discussion, lines 332 – 334).

Reviewer 1

1) The authors are able to show that the amount of IACSI was significantly reduced after implementation of the treatment algorithm (851:436) in a comparable period. Unfortunately, the reasons for the reduction of the number of interventions is not described. 

Author Response: We suggest that our QI investigation and recommendations (Figure 2) followed by the high-profile, institution-wide awareness campaign were responsible for the dramatic reduction in provider referrals for hip injections at our institution. We have added an explanation and clarification of this important point as follows:

• “…a high-profile, well-coordinated, institution-wide, campaign to raise awareness of work-group findings and recommendations was undertaken to educate providers in Radiology, Orthopedics, Rheumatology, Physical Medicine / Rehabilitation, and Primary Care. Guidelines and patient-education materials were then distributed hospital-wide for their use in clinical practice. Work-group findings and practice recommendations were also highlighted at our institution’s annual research symposium.” (Materials and Methods, lines 127 – 132). 

• “In the 27 months after QI investigation and implementation of practice recommendations (Figure 2), reduction in number of procedures was 48.8% (851 to 436), reduction in number of hips injected was 39.6% (531 to 321), number of injections per hip reduced from 1.61 to 1.37 (p = 0.0006), reduction in post-IACSI FHC was 67.9% (84 to 27), and rate of FHC decreased from 20.4% to 15.1% (p = 0.1292) (Table 3).” (Results, lines 186 – 190). 

2) It is not shown if there was a reduction of number of patients with the mentioned comorbidities in the second group.

Author Response: The reviewer makes a good point. The impact of our QI intervention on number of patients referred for hip IACSI with potential risk factors associated with FHC was not tracked or tabulated. This is a weakness of our study and has been acknowledged as below:

• “Although our initial QI investigation identified potential patient-related risk factors for FHC after IACSI and made recommendations for/against treatment based on these, the impact of our QI intervention on number of patients referred for hip IACSI with potential risk factors was not tracked or tabulated.” (Discussion, lines 324 – 327.)

3) Another major concern is the low rate of post-injection follow-up radiographs in the intervention group. Only 55.8% received a post-procedure x-ray of the hip. How did the authors exclude the development of FHC without follow-up radiographs? This is surprising, since the missing of a post-injection X-ray was an exclusion-criteria in the first group.

Author Response (part a): A goal of this study was to identify variation in practice of obtaining post-procedure X-rays of the injected hip before and after quality improvement (QI) intervention. For patients without post-procedure X-rays, we were unable to conclude if they experienced FHC, so they were excluded from the FHC analysis only. However, we included these patients in our analysis of radiographic follow-up and have reported rates of radiographic follow-up to illustrate variation in post-injection X-ray practices. We have clarified our exclusion criteria as below:

• “If a patient did not have a hip radiograph available at least one-month post-injection, the patient was excluded from FHC analysis, but was included in analysis of post-injection radiographic follow-up.” (Materials and Methods, lines 112-114)

Author Response (part b): While we cannot determine if patients without post-injection radiographs developed FHC during the study period, we can illustrate variation in practice of radiographic follow-up post-procedure. This point has been further clarified in the manuscript as below:

• “Variation in radiographic follow-up requirement persisted as x-rays were obtained for only 55.8% of patients post-procedure.” (Results, lines 191 – 192).

• “After QI intervention, we observed a significant decrease in number of repeat injections per patient, mean = 1.61 vs. 1.37, (p = 0.0006). However, variation persisted in follow-up practices and the number of patients with post-injection radiographs decreased to 55.8% from 77.6%.” (Discussion, lines 305 – 308).

• “Finally, a large number of patients (22.4% pre-intervention and 44.2% post-intervention) were excluded from final analysis of occurrence of FHC due to lack of post-injection imaging.” (Discussion, lines 332 – 334).

Author Response (part c): Even though the lack of radiographic follow-up precluded a complete analysis of the incidence of FHC after IACSI at out institution, the data collected was sufficiently concerning to warrant QI investigation and intervention to promote patient safety. This point is highlighted in the manuscript as follows: 

• “However, even if all patients initially excluded due to incomplete radiographic follow-up were included in the final analysis with the assumption of “no collapse”, FHC rates after IACSI remain high (i.e. pre-intervention = 15.8%; post-intervention = 8.4%). At our institution, even this “best case scenario” was sufficiently concerning to motivate further investigation and intervention.” (Discussion, lines 334– 338).

4. FHC is not always easy to detect in plane radiographs in patients with osteoarthritis of the same hip. In order to have objective results a group of independent specialists (radiologists or orthopedics) should have analyzed the radiographs.

Author Response: Pre- and post-injection radiographs were compared side-by-side with the criteria of obvious loss of sphericity of the femoral head as the definition of collapse. Each radiograph was reviewed by an orthopedic surgery resident and later reviewed by the senior author, an orthopedic surgery attending. In every case, FHC was dramatic, easily detected, and further verified by our institution’s official radiology reports. We have added an explanation and clarification of this important point as follows:

• “Each pre- and post-injection radiograph was compared side-by-side for interval changes. Post-injection FHC was defined radiographically as new loss of femoral head sphericity (Figure 1). Images were initially compared and interpreted by two orthopedic surgery residents and later independently verified by an attending orthopedic staff. In all cases, official radiology reports confirmed an interim change in femoral head sphericity as well.” (Materials and Methods, lines 115 – 119). 

Reviewer #2: 

This is the largest study of this kind, looking at the incidence of Femoral Head Collapse after Intraarticular Steroid Injection. Although retrospective in nature, this provides a basis for future study in this particular population. The data does support the conclusions. The statistical analysis was appropriate and rigorous. The data was available for review. The manner in which this manuscript is written flows well and is easily understandable.

Author Response: We appreciate the positive comments regarding our manuscript and thank the reviewer.

Reviewer #3: 

Thank you for your paper, it represents a hard and continuous effort to improve practice, though I think it may not represent a clear research topic rather than documenting your internal audit of your practice

Author Response: We appreciate the reviewer’s comments regarding our manuscript. This was a single center, before/after quasi-experimental study of our experience with hip injections. We have presented our results in the form of data and eye-witness experiences. We have added an explanation and clarification of this important point as follows:

• “This was a single center, before/after quasi-experimental study designed to bring about immediate quality improvements in health delivery in our institution. The Institutional Review Board provided exemption for this investigation and there was no external funding.” (Materials and Methods, lines 88 – 90).

---

## [Decision Letter · Decision Letter 1]

16 Aug 2021

PONE-D-21-00877R1

Femoral Head Collapse After Hip Intra-Articular Corticosteroid Injection: An Institutional Response to Improve Practice and Increase Patient Safety

PLOS ONE

Dear Dr. Sechriest,

Thank you for submitting your manuscript to PLOS ONE. After careful consideration, we feel that it has merit but does not fully meet PLOS ONE’s publication criteria as it currently stands. Therefore, we invite you to submit a revised version of the manuscript that addresses the points raised during the review process.

We look forward to receiving your revised manuscript.

Kind regards,

Osama Farouk

Academic Editor

PLOS ONE

Journal Requirements:

Reviewers' comments:

Reviewer's Responses to Questions

**Comments to the Author**

1. If the authors have adequately addressed your comments raised in a previous round of review and you feel that this manuscript is now acceptable for publication, you may indicate that here to bypass the “Comments to the Author” section, enter your conflict of interest statement in the “Confidential to Editor” section, and submit your "Accept" recommendation.

Reviewer #4: All comments have been addressed

Reviewer #5: All comments have been addressed

2. Is the manuscript technically sound, and do the data support the conclusions?

Reviewer #4: Yes

Reviewer #5: Yes

3. Has the statistical analysis been performed appropriately and rigorously? 

Reviewer #4: Yes

Reviewer #5: Yes

4. Have the authors made all data underlying the findings in their manuscript fully available?

Reviewer #4: Yes

Reviewer #5: No

5. Is the manuscript presented in an intelligible fashion and written in standard English?

Reviewer #4: Yes

Reviewer #5: Yes

6. Review Comments to the Author

Reviewer #4: Dear authors, I had the opportunity to review your work, and I hope that my comments will help improve the quality and strengthen your manuscript.

I encourage the authors to strictly follow the journal instructions; for example, it is not advised by the journal to insert the table or figure legends inside the manuscript.

Dear authors, I had the opportunity to review your work, and I hope that my comments will help improve the quality and strengthen your manuscript.

-The authors had responded rigorously to the comments raised after the initial submission; however, I have some minor comments

-line 93, the authors reported: “witnessing a high rate” how much are they considered high? Is there a certain percentage they calculated, or data were reported from a previous survey or study in their institution?

-line 116, the authors reported comparing the x-rays side by side; what was the exact tool used to judge the sphericity of the femoral head, how did they judge the proper rotation of the included radiographs? Did they exclude any cases due to bad-quality radiographs? Did the two residents performed checking of the whole x rays or these were split between them?

-line 189: “reduction in post-IACSI FHC was 67.9% (84 to 27)” was this reduction possibly due to the reduced number of patients who had an injection? Or due to a relatively shorter time of follow up after implementing the quality improvement protocol?

-I would like the authors to add a statement in the acknowledgment thanking the residents for their effort if they were not listed as authors.

Reviewer #5: Thank you so much for this interesting research topic.

-I wonder whether the inclusion of AVN cases ( 8 cases) and given the possibility that the natural progression of the disease rather than the IACSI was the cause of collapse rather than the IACSI??? I think this is worse elaborating upon specifically referring to the AVN cases please.

- Could you please elaborate more about the effect of the injection dose /material on the rate of collapse among those who have developed collapse and those who haven't please.

7. PLOS authors have the option to publish the peer review history of their article (what does this mean?). If published, this will include your full peer review and any attached files.

Reviewer #4: **Yes: **Ahmed A. Khalifa, M.D., FRCS, MSc.

Reviewer #5: **Yes: **Mahmoud Abdel Karim

---

## [Author Response · Author response to Decision Letter 1]

31 Aug 2021

Response to Reviewer Comments

PONE-D-21-00877R1

Reviewer 4: 

1) Line 93, the authors reported: “witnessing a high rate” how much are they considered high? Is there a certain percentage they calculated, or data were reported from a previous survey or study in their institution?

Author Response: 

At the end of 2017, we had observed and treated multiple patients who returned to clinic after hip intra-articular corticosteroid injection (IACSI) with debilitating symptoms secondary to femoral head collapse (FHC). Prior to our investigation, there was no known or previously reported rate of FHC at our institution. Additionally, there is a paucity of cases documented in the literature, and available case reports and series do not define any true incidence. However, FHC is considered a rare complication after IACSI. We have revised our manuscript as follows:

• “After witnessing multiple cases of FHC after hip IACSI, we reviewed and compared our experience and institutional care practices with the evidence-based literature.” (Materials & Methods, lines 93 - 94)

Reviewer 4: 

2) Line 116, the authors reported comparing the x-rays side by side; what was the exact tool used to judge the sphericity of the femoral head, how did they judge the proper rotation of the included radiographs? Did they exclude any cases due to bad-quality radiographs? Did the two residents performed check of the whole x-rays or these were split between them? 

Author Response (part a): 

Each radiograph was reviewed using the picture archiving and communication system (PACS). Pre- and post-injection radiographs were compared side by side for loss of contour secondary to collapse. For each hip image, a circle tool was used to outline the contour of the subchondral bone-edge in the weightbearing portion of the femoral head. We have added an explanation and clarification of methodology as follows:

• “Post-injection FHC was defined radiographically as new loss of femoral head sphericity and was measured by comparing the contour of the weightbearing portion of the femoral head on each radiograph. For each hip image, a circle tool was used to outline the contour of the subchondral bone-edge in the weightbearing portion of the femoral head.” (Materials & Methods, lines 117 -121)

Author Response (part b): 

The authors examined pre- and post-hip IACSI anterior-posterior (AP) pelvis or hip radiographs. Standard AP pelvis and/or hip radiographs were obtained. Per radiology department protocol, radiographs were reviewed by a radiology technician in real-time to ensure correct position and rotation is obtained. If a radiograph was inadequate a new radiograph was obtained prior to releasing the radiograph to PACS. There were no radiographs excluded due to malrotation. This point is highlighted in the manuscript as follows: 

• “All radiographs were obtained using our institution’s standardized Radiology department protocol. No images were excluded for poor quality.” (Materials & Methods, lines 114-116)

Author Response (part c): 

Each patient was reviewed independently by one of two orthopedic surgery residents. All patient radiographs then underwent confirmatory review by the senior author. This point is clarified in the manuscript as follows: 

• “Images were initially compared and interpreted by two orthopedic surgery residents and later independently verified by an attending orthopedic staff. In all cases, FHC was dramatic, easily detected, and the interim change in femoral head sphericity secondary to collapse was confirmed by official radiology reports.” (Materials & Methods, lines 121-124)

Reviewer 4:

3) Line 189. “Reduction in post-IACSI FHC was 67.9% (84 to 27)” was this reduction possibly due to reduced number of patients who had an injection? Or due to a relatively shorter time of follow up after implementing the quality improvement protocol?

Author Response: 

The reviewer raises a very important point. We agree that, due to the shorter length of radiographic follow up for the post-intervention group, there is a possibility of bias impacting the results, falsely lowering the rate of FHC. While this is a weakness of our study, the possibility of a falsely low FHC rate in the post-intervention group strengthens our rationale for instituting the Clinical Pathway to standardize and study hip IACSI best practices. This weakness is acknowledged and addressed in the manuscript as follows: 

• “Due to the retrospective nature of this work, the radiographic follow-up for pre- and post-intervention study groups was non-standardized. Additionally, the time to chart/radiographic review was shorter for the post-intervention group, thereby introducing the potential for a falsely decreased FHC rate in the post-intervention group. While the shorter length of radiographic follow up for the post-intervention group and the possibility of a falsely low FHC rate in this group is a weakness of our study, it strengthens our rationale for instituting the Clinical Pathway to standardize hip IACSI practices so to allow for further prospective study.” (Discussion, lines 351-357)

Reviewer 4:

4) I would like the authors to add a statement in the acknowledgement thanking the residents for their effort if they were not listed as authors.

Author Response: 

The residents responsible for the chart review are included as first and second authors. 

Reviewer 5

1) I wonder whether the inclusion of AVN cases (8 cases) and given the possibility that the natural progression of the disease rather than the IACSI was the cause of the collapse rather than the IACSI??? I think this is worth elaborating upon specifically referring to the AVN cases please.

Author Response: 

Avascular necrosis (AVN) or similar pathologies contributed to some of the FHCs observed at our institution. As this review was limited by its retrospective nature, we studied all patients referred for hip IACSI, including the 8 hips with a diagnosis of AVN. We have reported the diagnoses documented for all patients referred for hip IACSI as we felt it important to include all patients to best determine associated risk factors for FHC. Per the reviewer’s suggestions, we have added to the manuscript with additional information on the AVN cases as follows: 

• “There were 8 hips (1.51%) that had a diagnosis of avascular necrosis (AVN). One patient was excluded due to pre-existing FHC. Of the remaining 7 hips, 2 hips (2.38%) developed FHC, and 5 hips (1.52%) did not develop collapse after hip IACSI.” (Results, lines 174 - 176) 

• “The natural history of avascular necrosis of the femoral head is FHC. In our analysis there were only 8 hips (1.51%) that had a diagnosis of AVN (7 ultimately included in analysis). Of the patients with AVN there was a 28.57% FHC rate (2/7) after IACSI. While there is a possibility AVN was underdiagnosed in our patient population, we were limited by the retrospective nature of the review and the information contained in the medical record at the time of retrieval. Due to the slightly higher FHC rate seen with this diagnosis and the natural history of the disease, we consider AVN as a relative contraindication for hip IACSI.” (Discussion, lines 279 - 285)

Reviewer 5

2) Could you please elaborate more about the effect of the injection dose/material on the rate of collapse among those who have developed collapse and those who haven’t please?

Author Response: 

The reviewer raises an excellent point. There is basic science evidence showing dose and volume-dependent chondrotoxicity with local anesthetics and corticosteroids. However, in our study, the variation in dosage and volume of hip injectate between providers was very substantial such that no association could be readily identified between FHC and the injectate. Ultimately, in response to this provider-to-provider variation and using the best available evidence to inform the safest medications and dosages for joint injections, the recommendation for the least toxic medications in the lowest effective dosage and volume were made. This point has been further clarified in the manuscript as below:

• “No significant association was identified between FHC and local anesthetic or corticosteroid dosage, volume, or medication type.” (Results, lines 188 - 190)

---

## [Decision Letter · Decision Letter 2]

18 Oct 2021

Femoral Head Collapse After Hip Intra-Articular Corticosteroid Injection: An Institutional Response to Improve Practice and Increase Patient Safety

PONE-D-21-00877R2

Dear Dr. Sechriest,

We’re pleased to inform you that your manuscript has been judged scientifically suitable for publication and will be formally accepted for publication once it meets all outstanding technical requirements.

Kind regards,

Osama Farouk

Academic Editor

PLOS ONE

Additional Editor Comments (optional):

Reviewers' comments:

Reviewer's Responses to Questions

**Comments to the Author**

1. If the authors have adequately addressed your comments raised in a previous round of review and you feel that this manuscript is now acceptable for publication, you may indicate that here to bypass the “Comments to the Author” section, enter your conflict of interest statement in the “Confidential to Editor” section, and submit your "Accept" recommendation.

Reviewer #4: All comments have been addressed

2. Is the manuscript technically sound, and do the data support the conclusions?

Reviewer #4: Yes

3. Has the statistical analysis been performed appropriately and rigorously? 

Reviewer #4: Yes

4. Have the authors made all data underlying the findings in their manuscript fully available?

Reviewer #4: Yes

5. Is the manuscript presented in an intelligible fashion and written in standard English?

Reviewer #4: Yes

6. Review Comments to the Author

Reviewer #4: Dear authors, i was glad to review the revised version of your manuscript. Thanks for addressing all the raised issues, hope you all the best.

7. PLOS authors have the option to publish the peer review history of their article (what does this mean?). If published, this will include your full peer review and any attached files.

Reviewer #4: **Yes: **Ahmed A. Khalifa, M.D., FRCS, MSc

---

## [Editor Report · Acceptance letter]

21 Oct 2021

PONE-D-21-00877R2 

Femoral Head Collapse After Hip Intra-Articular Corticosteroid Injection: An Institutional Response to Improve Practice and Increase Patient Safety 

Dear Dr. Sechriest:

I'm pleased to inform you that your manuscript has been deemed suitable for publication in PLOS ONE. Congratulations! Your manuscript is now with our production department. 

Kind regards, 

on behalf of

Dr. Osama Farouk 

Academic Editor

PLOS ONE